# Beyond the Alphabet: Deep Signal Embedding for Enhanced DNA Clustering

## Abstract

The emerging field of DNA storage employs strands of DNA bases (A/T/C/G) as a storage medium for digital information to enable massive density and durability. The DNA storage pipeline includes: (1) encoding the raw data into sequences of DNA bases; (2) synthesizing the sequences as DNA *strands* that are stored over time as an unordered set; (3) sequencing the DNA strands to generate DNA *reads*; and (4) deducing the original data. The DNA synthesis and sequencing stages each generate several independent error-prone duplicates of each strand which are then utilized in the final stage to reconstruct the best estimate for the original strand. Specifically, the reads are first *clustered* into groups likely originating from the same strand (based on their similarity to each other), and then each group approximates the strand that led to the reads of that group. This work improves the DNA clustering stage by embedding it as part of the DNA sequencing. Traditional DNA storage solutions begin after the DNA sequencing process generates discrete DNA reads (A/T/C/G), yet we identify that there is untapped potential in using the raw signals generated by the Nanopore DNA sequencing machine before they are discretized into bases, a process known as *basecalling*, which is done using a deep neural network. We propose a deep neural network that clusters these signals directly, demonstrating superior accuracy, and reduced computation times compared to current approaches that cluster after basecalling.

## 1 Introduction

The rapid growth of digital data, projected to reach 180 zettabytes by 2025, is causing a data storage crisis that cannot be addressed by existing storage technologies (Rydning, 2022). In response, deoxyribonucleic acid (DNA) is emerging as a promising alternative storage medium due to its incredible density and durability. The DNA storage process includes four stages illustrated in Figure 1: (1) an *"encoding"* stage in which binary data files are encoded into DNA strands (design files) using error-correcting code (ECC) (Koblitz et al., 2000) schemes that may also overcome errors, (2) a *"synthesis"* stage, which produces artificial DNA strands of each design strand and are then stored in a *storage container* (LeProust et al., 2010), (3) a *"sequencing"* stage (Anavy et al., 2019; Erlich & Zielinski, 2017; Organick et al., 2018; Yazdi et al., 2017) which translates a DNA strand into a digital sequence known as a *"read"*, and (4) a *"retrieval"* stage where reads are decoded back to binary data files while correcting any errors using the chosen coding methods. Despite the vast potential of DNA storage, current DNA sequencers are yet to overcome challenges such as low throughput and high costs compared to the traditional alternatives (Alliance, 2021; Shomorony et al., 2022; Yazdi et al., 2015).

The emerging Nanopore technology offers real-time sequencing of DNA strands with drastically lower costs and portability compared to traditional Illumina sequencing machines (Jain et al., 2016; Kono & Arakawa, 2019). Despite having higher error rates compared to other sequencing technologies such as Illumina, Nanopore sequencing is gaining significant attention due to its lower cost, portability, and capability to sequence longer strands of DNA. Nanopore sequencing directly reads the nucleic acids by passing them through a nanoscale pore, called a nanopore, embedded in a membrane. As the nucleic acid strand moves through the pore, it impacts the electrical current in a fashion dependent on the current bases in the pore (between four and six at any given time) (Mao et al., 2018; McBain & Viterbo, 2024). The current produced by the DNA molecules passing through the pore is converted into raw analog signals. In Nanopore sequencing, basecalling entails converting these raw signals into

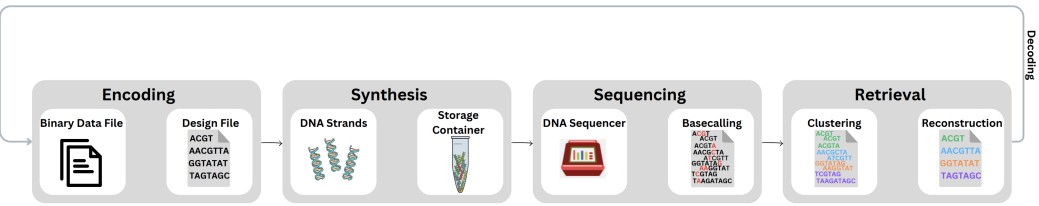

Figure 1: The stages in the DNA storage pipeline.

nucleotide sequences composed of the bases $\{A, T, C, G\}$ (McBain & Viterbo, 2024). Basecalling is done using deep neural networks.

While Nanopore sequencing benefits from a drastic reduction in costs, it also poses several challenges due to significantly higher sequencing error rates (approximately 10% of bases suffer from edit, insertion or deletion errors as compared to only 1% in Illumina) (Chandak et al., 2020). This paper overcomes these errors by exploiting the raw Nanopore signals which contain significantly more information as compared to the discrete post-basecaller read. Unfortunately, direct utilization of the raw signal is challenging since, to the best of our knowledge, no previous work in DNA storage has directly exploited the signals for processes such as clustering, reconstruction, and error correction (Chandak et al., 2020).

A major stage in the DNA storage pipeline involves clustering billions of strings based on the edit distance metric in order to identify which groups of reads were likely duplicated from the same source DNA strand. The edit distance between two strings $x, y$ denoted as $d_e(x, y)$ is the minimum number of edits, deletions, or insertions required to transform $x$ to $y$ (Ristad & Yianilos, 1998). The complexity of this procedure is caused by several factors: (1) the variability and probabilistic nature of the cluster size due to varying duplicate counts; (2) the number of original clusters in the design files; (3) the protocols used in the experiments, for example, PCR errors pre-sequencing procedure and sampling errors that occur during nanopore sequencing machine; and (4) the fact that the computation of a distance matrix based on the edit distance between each pair of reads is not scalable for typical DNA storage datasets that are very big.

Existing clustering algorithms often suffer from high computational time and compromised accuracy (Qu et al., 2022; Zorita et al., 2015). While approximation algorithms and tools such as metric embeddings and locality-sensitive hashing (LSH) offer promising avenues by sacrificing some accuracy for substantial reductions in runtime, their applicability to edit distances remains poorly understood. For the retrieval of a given file, the *sequencing* and *retrieval* stages take approximately a number of days to compute; therefore, any reduction in computation time is a major step forward.

At the heart of the challenges of clustering DNA strands for DNA storage lies the impracticality of computing the edit distance between every pair of input strands to create a distance matrix. Due to the sheer volume of data, this is exacerbated by the quadratic time complexity of edit-distance algorithms. Our work addresses this concern by proposing a novel approach: searching for similarity directly in the raw analog DNA signals rather than the discrete DNA reads. Our work introduces a paradigm shift in the field of DNA data storage retrieval by leveraging deep neural network embedding directly on the raw signals instead of the post-basecaller strands. By embedding the raw signals directly in an architecture composed of convolutional neural network (CNN) layers, long-short-term memory layers (LSTM), and linear layers, we bypass the limitations associated with clustering based on the A/T/C/G alphabet, allowing us to cluster raw signals before passing them through the basecaller. This step can significantly reduce the errors introduced during the sequencing phase and lead to a more accurate reconstruction of the original design file.

The key contributions to our work are as follows:

- We are the first to perform similarity groupings of DNA strands directly on the raw electrical signals.

- The proposed approach significantly reduces sample exclusion because it uses the continuous dimension of electrical signals, which reduces the loss of information during the conversion to the discrete dimension after basecalling.

- The proposed approach demonstrates the ability to achieve between 1 to 3 order-of-magnitude faster computation times while producing clusters with high enough quality to allow data reconstruction.

The rest of this paper is organized as follows: Section 2 defines the problem, and Section 3 reviews related work. Section 4 introduces the uniqueness of the datasets and their creation, Section 5 describes the proposed deep learning model, and Section 6 presents an evaluation of the trained ML models across different experiments. Section 7 uses the trained ML models for applications in the DNA storage pipeline, and Section 8 concludes the paper.

## 2 PROBLEM STATEMENT

The main objective of this work is to develop a distance function for raw DNA signals that effectively captures the similarity between them. Thus, the goal is to ensure that raw signals that originate from the same cluster are close together in the metric space, while those that are not are far apart. We formulate the problem as a supervised learning problem, where signals belonging to the same cluster should have small distances between them in the transformed feature space, while signals from different clusters should have large distances between them in the transformed feature space. More specifically, let, without loss of generality, $C = \{1, \ldots, m\}$ be the set of $m$ labels originating from the design file and let $S = \{s_1, s_2, \ldots, s_n\}$ be the set of raw DNA signals, each belonging to one of the $m$ labels (we denote the label of $s_i$ by $c_i$). Our goal is to find a distance function $D$ that is targeted to achieve the following properties:

1. **Similarity Preservation**: If two samples $s_i$ and $s_j$ belong to the same class ($c_i = c_j$), the distance $D(s_i, s_j)$ should be minimized.

2. **Dissimilarity Preservation**: If two samples $s_i$ and $s_j$ belong to different classes ($c_i \neq c_j$), the distance $D(s_i, s_j)$ should be maximized.

3. **Robustness to Noise**: The distance function should be robust to noise and variability inherent in raw DNA signal data.

4. **Computational Efficiency**: The metric should be computationally feasible for large datasets, where $D(s_i, s_j)$ computation has to be done in $O(1)$.

## 3 RELATED WORK

The clustering phase of the DNA storage pipeline has been explored solely in the context of clustering the discrete DNA reads. The number of clusters in the clustering phase of the DNA storage pipeline is typically between $1,000$ and $100,000$, while each cluster contains between 10 and $1,000$ samples. To cluster DNA strands, the similarity between pairs of reads is computed – typically using the edit-distance method. This presents two main challenges: (1) the computational complexity of pairwise comparisons being $O(n^2)$ where $n$ is the number of reads, and (2) the $O(\ell^2)$ complexity of the edit-distance computation, where $\ell$ represents the reads length. DNA clustering algorithms address these challenges by employing (1) *filtering* techniques to reduce the number of pairwise comparisons and (2) *approximation* methods to expedite edit-distance calculations. Both approaches aim to decrease the time complexity with minimal accuracy loss.

Several DNA clustering algorithms in bioinformatics and metagenomics, such as UCLUST (LeProust et al., 2010), CD-HIT (Fu et al., 2012), and USEARCH (Srinivasavaradhan et al., 2021) (which UCLUST is based on), use different methods to deal with the two challenges. One approach involves using greedy methods for filtering, such as identifying common short sub-sequences as a preliminary step before edit-distance computation. Location Sensitive Hashing (LSH), which is used in (Antkowiak et al., 2020; Rashtchian et al., 2017; Ben Shabat et al., 2023), is another emerging method that limits the edit-distance calculation to strands that share the same LSH value to make filtering more efficient.

To address the latter challenge, the edit-distance computation, several methods have been proposed. First, one can restrict the computation depth of the edit-distance, as demonstrated in (Bao et al., 2011), to mitigate the complexity. Second, the edit-distance can be calculated based on fixed-sized short sub-strings, as implemented in Starcode (Zorita et al., 2015) using short DNA bar-codes. Another approach leverages heuristic similarity instead of edit-distance to expedite processing times, as demonstrated in Clover (Qu et al., 2022) utilizing a tree structure.

We acknowledge several works (Ji et al., 2021; Li et al., 2021) that have been done in the context of DNA embeddings. Those works differ from ours since they involve correcting strands to align with a protein or gene. It is a process that is irrelevant in our case because our data is synthetic, and we aim to reconstruct our original data without, for example, capturing protein structure.

In the signal domain, a common method for comparing signals is Dynamic Time Warping (DTW) (Senin, 2008). DTW is a technique designed to compare two time series data sequences with varying lengths whose data points are not aligned, much like strands in the discrete domain, allowing for insertions, deletions, and substitutions. Similar to edit-distance, DTW's computational complexity is $O(m^2)$, where $m$ represents the signal length. Given that signal lengths are typically ten times longer than DNA strands, DTW computation becomes impractical for large datasets that are used in DNA storage applications. The focus of this work is in the signal domain; therefore, it is imperative to first develop an accurate solution to the second challenge.

## 4 DATASETS

Public DNA storage datasets do not share the raw signals, recall that during during the DNA storage pipeline (Figure 1), the mapping between the DNA strands and their corresponding reads involves an intermediate stage where the DNA strands are converted to raw signals. Therefore, it is necessary to create our own datasets from real experiments that maps raw signals to their corresponding DNA reads.

Each experiment in the DNA storage domain requires choosing an error-correcting code as part of the *"encoding"* stage for file reconstruction. In addition, a design file is generated for each experiment. The end-to-end experiment process involves DNA strand *"synthesis"* stage, followed by a *"sequencing"* stage, the DNA sequencer process approximately lasts 72 hours. Then basecalling is performed on the DNA reads, which takes approximately a week. Lastly, creating the ground truth for the dataset lasts between $6 - 30$ days.

Table 1 shows three design files and their corresponding Nanopore sequencing experiments [1]. The table outlines three different experiments that vary the number of clusters, the length of the DNA strands, and the average edit-distance between strands in the design files, therefore resulting in high variability in key parameters for the proposed datasets.

The design files of the different experiments encompass encoded data, index information for orderly reconstruction, and a consistent primer shared among all reads, positioned at the beginning and end of each strand within the file. Subsequently, the files were synthesized by "Twist Bioscience", resulting in "fast5" (Payne et al., 2019) file formats. The raw signal data is encapsulated in "fast5" file formats, which are then processed by a basecaller to convert it into "fastq" format. The "fastq" format represents the sequenced data using the following symbols: $\{A, T, C, G\}$. Each "fast5" file is linked to a "fastq" file, with a "read-id" serving as the connecting identifier. Given that the standard signal length in DNA storage datasets typically reaches up to 3000 units, for 95% of the cases, we uniformly reshaped all of the signals to this length using common techniques.

The ground truth for clustering is obtained by brute-force calculating the edit distance between each DNA read produced by the basecaller and the original strands from the respective design file. We then assign each DNA read to the cluster corresponding to the closest DNA strand from the design file. It is not practical or scalable to perform a full edit-distance computation due to its quadratic time complexity between the millions of reads per experiment, and their corresponding design files. To solve this, we impose a $k = 20$ limit constraint, where $k$ is the maximum distance between any

---

[1] The "Microsoft Experiment" design file is taken from (Srinivasavaradhan et al., 2021) and the sequenced data is from (Sabary et al., 2024). The "Deep DNA test" and "Deep DNA pilot" design file and sequenced data are from (Bar-Lev et al., 2021).

Table 1: Experiments differences

| Experiment | No. Clusters | Strand Length | Avg Edit Dist. | Avg No. Copies | Avg Strand Length post Basecalling | Time Length Creation |
|---|---|---|---|---|---|---|
| Microsoft Experiment | 10, 000 | 110 | 52.6376 | 300 | 211 | 18 days |
| Deep DNA Pilot | 1, 000 | 140 | 64.4932 | 1, 300 | 240 | 6 days |
| Deep DNA Test | 110, 000 | 140 | 4.0328 | 10 | 243 | 30 days |

two strands, and all possible edit values between two strands will be within the range of $[0, k]$. This is a common approach used in extensive edit-distance computations. Table 1 shows that for each of the experiments, $k = 20$ ensures each strand will be categorized to the correct cluster with high probability since the majority of the reads are closely aligned to their intended design.

## 5 PROPOSED DEEP LEARNING SCHEME

We propose a deep-learning-based algorithm for computing similarity measures between the raw signals of DNA reads. This is done by leveraging the Dorado basecaller (Technologies, 2024) pre-trained weights. The architecture extends the Dorado model's feature extractor, coupled with the ArcFace loss function. In the following section, we explain our solution in detail.

The neural network (NN) architecture begins with the Dorado basecaller (Technologies, 2024), a key component from the Oxford Nanopore Technologies (ONT) product suite, designed for DNA sequencing via nanopore technology. ONT has developed several basecalling models that leverage deep learning (DL) to translate the raw electrical signals from nanopore sequencing into nucleotide sequences. These DL-based basecallers have significantly improved the accuracy and speed of nanopore sequencing data analysis.

The Conditional Random Field (CRF) layer in the Dorado model is employed as a feature extractor for encoding raw signals within the NN architecture. The CRF architecture includes several components, each with its specific role in processing the sequencing data. Initially, convolutional layers are deployed to extract salient features from the raw input signals through a sequence of convolutional operations, capturing critical patterns essential for the basecalling task. Following are recurrent layers (LSTM) that capture the sequential nature of the DNA sequence. They process the outputs from the convolutional layers, taking into account the order and dependencies between the nucleotides. The data is then passed through linear layers, which perform a linear transformation to align the processed features with the final output space. This is followed by a Clmap layer that ensures that the scores used for predicting the DNA sequence are within a specified range. This entire sequence of operation is called the CRFModel.

Dorado is an open-source basecaller for Oxford Nanopore reads (Technologies, 2024). We utilize the initial pre-trained layers of this model, truncating its architecture after the CRF layer, which outputs a size of 1024. Subsequently, we append a linear layer with dimensions $1024 \times 500$, where 500 represents the number of classes. The training process is coupled with the ArcFace loss function (Deng et al., 2019). This function is used in facial recognition and other DL tasks that involve distinguishing between different classes. It is designed to enhance the discriminative power of the feature embeddings produced by a NN. ArcFace loss achieves this by enforcing a margin between the features of different classes in the angular space, making the decision boundary more distinct. Specifically, it adds an angular margin penalty to the target angles in the softmax function, which forces the NN to learn more separable features (Khan et al., 2024). The loss uses a cosine similarity in its implementation. This results in a more compact clustering of features from the same class and a wider separation between features of different classes. This loss function assists deeply in quantifying the similarity between the different signals.

During training, we employed a fine-tuning approach by initializing our model weights with pre-trained weights of the Dorado model and then iteratively updating them during training. For training purpose, 500 clusters consisting of $205, 690$ samples were selected arbitrarily from the "Microsoft Experiment" dataset whose size is in the magnitude of millions of samples. The samples are divided into three sets. The first contains 70% of the samples, and it is used for training. The second contains 15% of the samples, and it is used for validation, and the remaining 15% are used for testing purposes, and to visualize the test set embedding. The ML models are trained with a batch size of 256 samples using the Adam optimizer (Kingma & Ba, 2014). During the first 50 epochs of training, the training was conducted on a subset of 100 clusters, from the initial selected 500. For the remaining 50 epochs,

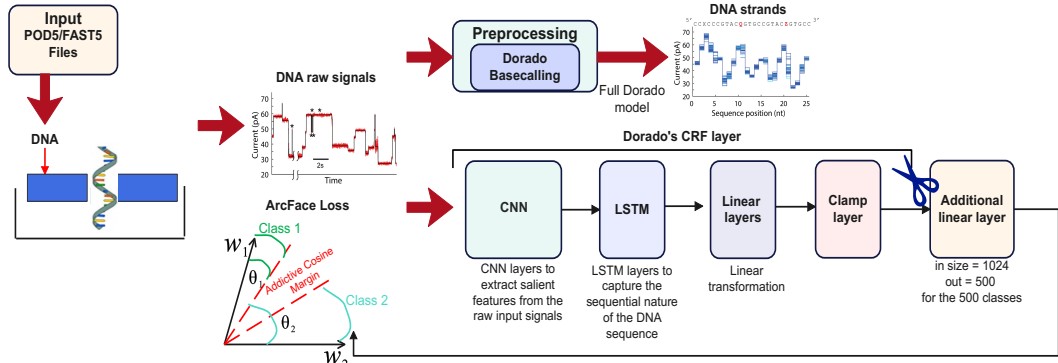

Figure 2: Deep signal embedding scheme, using a trimmed version of the Dorado model with an appended linear layer, using the ArcFace Loss function

training was conducted on all the clusters. When training on a large amount of data, using the ArcFace Loss, a common approach is to divide the training epochs, by gradually increasing the size of the training set (Khan et al., 2024). The performance is measured on a 2x AMD EPYC 7513 (32 core) server with 1TB RAM and four NVIDIA A6000 GPUs.

To evaluate the model, t-Distributed Stochastic Neighbor Embedding (tSNE) is used on the output's embeddings of the model, on the test set. tSNE is a dimensionality reduction technique utilized for visualizing high-dimensional data in lower-dimensional spaces, with an emphasis on preserving the local and global structure of the data points. Figure 3 illustrates the visualization of tSNE on the test set, demonstrating a distinct separation between the different clusters.

## 6 RESULTS

For each of the three datasets created in this work, we have the raw signals and the sequenced bases for each DNA read. Each of these datasets contains 50 clusters. The "Deep DNA Pilot" dataset's size is $63,849$ samples, the "Deep DNA Test" dataset's size is 739 samples, and the "Microsoft Experiment" dataset's size is $16,109$ samples that were not used during the training phase. These datasets ensure a diverse evaluation. For each of the three experiments, a binary ground truth square matrix was constructed with a set of 50 clusters. In this matrix, the $[i, j]$ entry equals to 1 only if the $i$ and $j$ signals belong to the same cluster.

The results present a comparison between raw signals' embeddings using a cosine similarity, and DNA strands using the edit-distance similarity. Recall that when using edit-distance, the possible values are amongst $\{0, 1, \ldots, k\}$, where $k$ is the upper limit of insertion, deletion or substitution. For each different $k$ value, an edit-distance similarity matrix is calculated, where a higher score entails a higher dissimilarity. The $k$ values chosen are 10, 20, 50, and 200; For example, for $k = 10$, the tested thresholds set is $\{0, 1, \ldots, 10\}$, and for each threshold a prediction matrix is calculated. These values exemplify the typical approach taken by approximation-

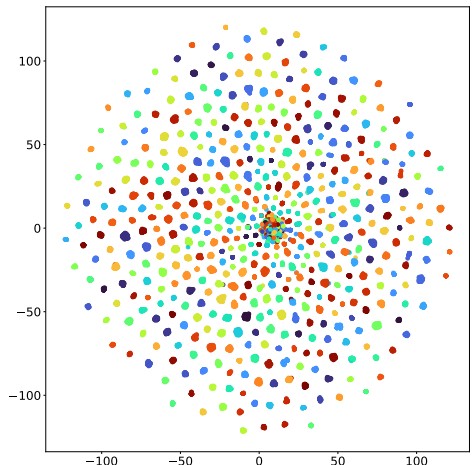

Figure 3: The test set is visualized using tSNE, with each color representing a distinct cluster. The presence of 500 clusters makes it difficult to distinguish similar colors visually. Each dot-like shape in the Figure represents a single cluster that is primarily well-separated from the other clusters.

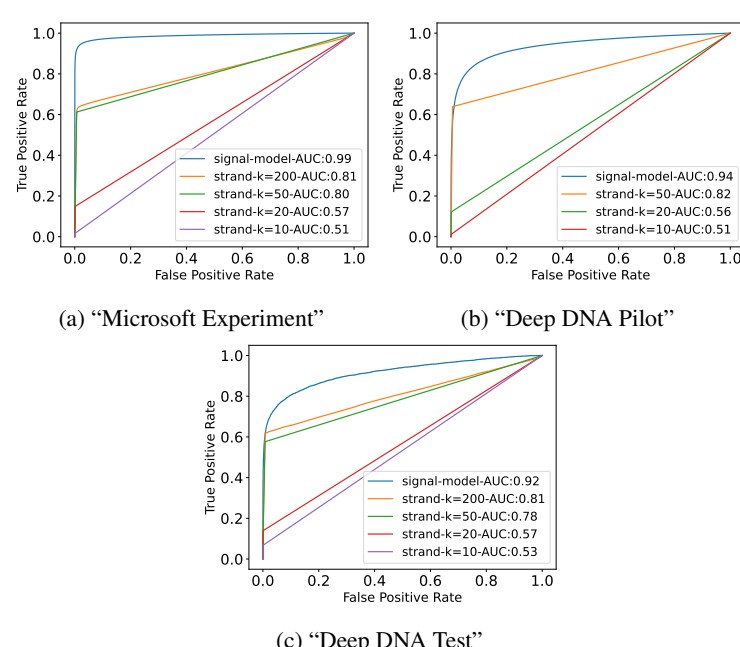

Figure 4: Signal similarity vs. sequence similarity across the different experiments

based DNA clustering algorithms to achieve faster computation times with minimal loss of accuracy. Such algorithms often use constraints or reduction techniques in edit-distance calculations (Starcode (Zorita et al., 2015), Meshclust (James et al., 2018), SEED (Bao et al., 2011) and Microsoft algorithm (Rashtchian et al., 2017)).

Every non-heuristic-based DNA clustering algorithm that measures similarity without direct computation relies on an edit-distance constraint. The ArcFace Loss (Deng et al., 2019) uses a cosine similarity for separating between inputs, therefore, a cosine similarity matrix is created to compare every pair of signals. For cosine similarity, the thresholds are continuous and not discrete as when using edit-distance, thus, containing much more data points, ranging between $-1$ and $1$, where $1$ is identical. Using these matrices, receiver operating characteristic (ROC) are generated. The ROC curve illustrates the trade-off between the true positive (TP) and false positive (FP) values of an machine learning model.

Figure 4 shows the ROC curves for the proposed signal-model and the three different baselines that use edit-distance similarity of the DNA strands with varying values of $k$. We expect that models with better classification performance will have ROC curves closer to the top-left corner since this indicates high true positive rate with a small false positive. The curve of a random ML model is expected to be close to the diagonal identity line. We quantitatively measure this performance according to the area under the ROC curve (AUC) metric, normalized to $[0, 1]$, where $1$ indicates perfect classification.

Figure 4 presents the ROC curve results for the three different experiments. The figure shows that the model that uses raw signals as inputs (signal-model) outperforms the models that use DNA strands as inputs (strand-$k = xx$) as higher TP rates are achieved for the same FP rates. Quantitatively, for the "Microsoft Experiment", we find that the signal-model achieves the best trade-off between TP and FP with an AUC of $0.99$ as compared to $0.81$, $0.80$, $0.57$, and $0.51$ for the strand-$k = 200$, the strand-$k = 50$, the strand-$k = 20$, and the strand-$k = 10$ models, respectively. The sharp turn and linear increase across all the models, after the rapid rise, is attributed to read pairs with distance above $k$, thus, they are all treated as equally distant. The strand-$k = 200$ is the only model that is comparable performance wise to the signal-model, though its complexity is much higher. Recall from Table 1, the average strand's length is approximately 200, therefore using an edit-distance with a $k = 200$ value, can be treated as an unbounded edit-distance calculation. Similar trends can be seen for "Deep DNA Pilot" and "Deep DNA Test" (refer to Figures 4(b) and 4(c), respectively). Notice that the superior performance in the "Microsoft Experiment" compared to the other experiments is

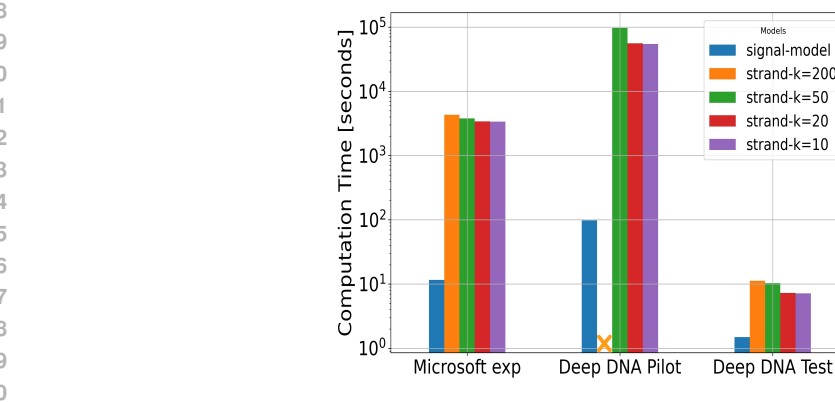

Figure 5: Compute times for three different experiments. For the signal-model it includes both the embeddings and the cosine similarity computation times; for the different strand-$k = xx$-models it includes edit-distance similarity matrix computation time. The red cross indicates the omitted entries in which the running time is larger than 30 hours

due to the training data originating from the same dataset. Even though the clusters that are used for evaluation were not present during the training process, the training process was conducted on the same dataset and thus the signals are taken from the same data distribution.

An important property of our similarity scheme is its execution time, as shown in Figure 5 on a logarithmic scale. It shows the computation time for the similarity matrices for each model, across all experiments. For the strand-$k = xx$ models, this entails the edit-distance similarity matrix computation time, and for the signal-model it entails the feed-forward that generates the embedding, and the cosine similarity calculation times. For the "Deep DNA Pilot" dataset, the computation time of the strand-$k = 200$-model is marked using a red cross, because its computation time is larger than 30 hours.

It is evident that the running time of strand-$k = xx$ models is between 1 and 3 orders of magnitude larger than that of signal-model. It is also evident that strand-$k = xx$ models have a linear increase between the different $k$ values across the different experiments. The variation between the datasets is due to is due to the varying dataset sizes applied to the same model.

## 7    APPLICATIONS

Dealing directly with the raw signals has the potential to revolutionize the pipeline for analyzing Nanopore sequencing, especially in the context of DNA storage. Two critical aspects of the DNA storage pipeline are the clustering and reconstruction phases, which could benefit significantly from this approach.

### 7.1    CLUSTERING PHASE

To show the potential of the signal-model in the clustering phase, we used a hierarchical clustering algorithm (Murtagh & Contreras, 2012) that is compatible with cosine similarity. We conduct the evaluation on the three datasets introduced earlier. Unlike modern DNA clustering algorithms which are commonly used, such as Clover (Qu et al., 2022) and Microsoft's (Rashtchian et al., 2017), the signal-model computation time is considerably faster. Additionally, we include all the raw signals samples, unlike for example Clover, which excludes 3% of the strands samples. We compare the proposed signal-model to Clover, since Clover's execution time is the only one that is comparable in its execution time, however Clover's accuracy is significantly lower than the signal model's. The number of clusters that Clover returns is different from the actual number, 50, and the number of samples in each cluster greatly differs from the ground truth. Microsoft's algorithm (Rashtchian et al., 2017) is orders of magnitude slower than Clover; see e.g. (Ben Shabat et al., 2023; Qu et al., 2022), it also discards a notable number of reads, and similar to Clover, the number of clusters it produces does not align with the original cluster count, though its accuracy is higher. Both, Clover and Microsoft's

Table 2: Clustering scores for the three experiments using raw DNA signals

| Dataset | Time [s] | Rand | Homogeneity | Completeness | V measure |
|---|---|---|---|---|---|
| Microsoft Experiment | 58 | 0.971 | 0.790 | 0.959 | 0.866 |
| Deep DNA Pilot | 970.82 | 0.887 | 0.528 | 0.859 | 0.654 |
| Deep DNA Test | 0.12 | 0.978 | 0.821 | 0.857 | 0.839 |

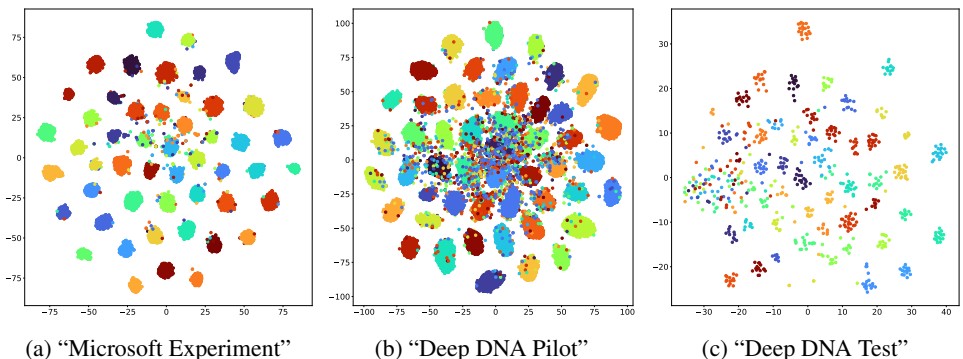

(a) "Microsoft Experiment"  (b) "Deep DNA Pilot"  (c) "Deep DNA Test"

Figure 6: TSNE Analysis of 50 Clusters from Various Experiments

algorithm, output a subset of highly distinguished clusters, while excluding many samples. Unlike them, Signal-model outputs clusters which are distinguishable enough so reconstruction can be done, across all clusters.

Table 2 presents time and clustering scores for the three out-of-sample datasets introduced earlier, for the signal-model. It should be noted that unlike the traditional DNA strands clustering, the signal-model achieves its results on the complete set of original clusters and not a partial set. Recall that the datasets differ in size in orders of magnitude, where the "Deep DNA pilot" is the largest dataset, followed by the "Microsoft" dataset, and lastly, the "Deep DNA Test".

The Table lists four different scoring measures: (1) The Rand score measures the similarity between two clusterings, which are different ways of grouping a set of data points into clusters, by evaluating the proportion of samples pairs that are either placed in the same cluster or placed in different clusters. It ranges between 0 and 1, where a score of 1 indicates that the clusters are exactly identical, and a score close to 0 suggests almost no agreement between the clusters. As the table reports, across all three datasets signal-model achieves a very high accuracy above 88%; (2) The Homogeneity score assesses clustering quality by determining if each cluster exclusively contains members of a single class. It ranges between 0 and 1, where a score of 1 indicates perfect homogeneity with each cluster composed entirely of samples from a single class, whereas a score closer to 0 indicates that clusters contain a mix of different classes. It is evident from the table that using a larger dataset, it hinders the performance; (3) The Completeness score evaluates clustering quality by checking if all samples of a given class are assigned to the same cluster. It ranges between 0 and 1, with a score of 1 indicating perfect completeness, meaning that all data points belonging to a given class are entirely within a single cluster. Across the three datasets, the performance of the signal-model is consistently high and comparable; and (4) the V measure score, which is a harmonic mean of homogeneity and completeness scores. It provides a single measure to assess the quality of a clustering output. It ranges between 0 and 1, where 1 indicates perfect clustering with maximum homogeneity and completeness. Signal's model performance is very high for "Deep DNA Test" and Microsoft datasets, while for "Deep DNA pilot" the performance is worse.

Figure 6 shows the tSNE visualizations of the signal-model across the three datasets. Figure 6(a) shows that the "Microsoft Experiment" has the best separation among the clusters, as explained earlier. Figure 6(b) exhibits higher density with larger clusters, while Figure 6(c) appears sparse with smaller clusters, as outlined in Table 1.

## 7.2 RECONSTRUCTION PHASE

The reconstruction phase is done by taking the output clusters produced by the models or algorithms (Pan et al., 2022; Sabary et al., 2024) and it computes the error rate within each cluster. Then, the algorithms selectively remove reads to achieve a specific error rate conducive to successful reconstruction of the design file. In contrast, since signal-model directly uses the raw signals, the clustering algorithm will be given more data for the reconstruction algorithms. This is because signal-model's approach significantly reduces the sample exclusions, but still outputs well enough distinguishable clusters. This enables the removal of reads as necessary, potentially yielding equivalent or improved results. When certain clustering algorithms fail to produce all clusters, signal-model ensures that all clusters are available for reconstruction.

## 8 DISCUSSION AND CONCLUSION

Directly utilizing raw DNA signals for analyzing Nanopore sequencing data offers a transformative approach to the DNA storage pipeline, particularly in the clustering and reconstruction phases. The signal-based approach enhances both clustering and reconstruction, showing superior computational efficiency and accuracy over traditional DNA strand-based clusterings. Experiments reveal that the signal-model outperforms conventional clustering algorithms like Clover and Microsoft's, providing faster and more accurate results. Additionally, the signal-model demonstrates significantly lower computation times, by orders of magnitude, highlighting its efficiency and scalability for large-scale DNA storage tasks. This paper advocates for a shift towards using raw DNA signal data, improving current methodologies, and paving the way for future advancements in DNA storage and analysis technologies.

This paper ignites research on using raw DNA signals directly, prompting the development of a clustering algorithm suitable for raw DNA signals. Future work can extend the approach of using the raw signals to enhance the reconstruction algorithms as well as the decoding procedure of error-correcting codes in the DNA storage pipeline. Additionally, further extensions of this methodology could apply to DNA and RNA sequencing across various fields, including bioinformatics, chemistry, and biology.

**Limitations:** The approach is tailored for Nanopore sequencing, but any modifications to the Nanopore machine may necessitate retraining the ML model, increasing the complexity of its practical application. While effective with up to 500 clusters, the algorithm has not yet been scaled for thousands of clusters, which could pose additional challenges. The implementation relies on the open-source Dorado model (Technologies, 2024) weights, introducing dependencies on external updates. Additionally, further adjustments may be needed to adapt the algorithm for biological tasks, particularly signal comparison, due to challenges like varied-length reads, which differ from the DNA storage context.

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
