# OpenReview forum: "Beyond the Alphabet: Deep Signal Embedding for Enhanced DNA Clustering"
_ICLR.cc/2025/Conference — ICLR 2025 Conference Withdrawn Submission_

### Official Review · Reviewer_ErVP · 2024-10-27

**Soundness:** 2
**Presentation:** 2
**Contribution:** 2
**Rating:** 3
**Confidence:** 4

**Summary:**

The manuscript introduced a DL-based approach for clustering sequences retrieved from the DNA sequencing pipeline, utilizing raw signals from Nanopore sequencing method.  The proposed model is derived through fine-tuning an open-source signal-to-base model known as Dorado, under a classification task.

**Strengths:**

The approach of commencing the analysis from raw Nanopore signals, rather than relying on pre-processed discrete DNA sequences, represents a novel direction in the field.

Sequence clustering is a challenging problem, especially when dealing with a large number of sequences. The proposed method may have significant impact on the DNA storage community.

**Weaknesses:**

There are several weaknesses that the authors may need to address.

0. The authors should pay more attention to the typos, grammars, etc. e.g line125 ", while those that are not are for apart. "

1. There are existing works that employs deep learning techniques to assess the similarity between sequences (DNA sequences), which are very closely related to this manuscript but omitted in this work. e.g.
```
1. "Convolutional embedding for edit distance."  SIGIR 2020
2. "Neural distance embeddings for biological sequences."  NeurIPS 2021
3. "Deep Squared Euclidean Approximation to the Levenshtein Distance for DNA Storage." ICML 2022
```

2. To the best of the reviewer’s knowledge, none of the three datasets utilized in this study are available in an open-access format as raw signals. This limitation may impede the reproducibility and validation of the proposed method.

3. The author(s) have chosen to employ the ArcFace loss function as a substitute for the cross-entropy loss in the classification task. However, the manuscript lacks an analysis or ablation study that would substantiate the necessity of introducing this more complex loss function.

4. The contribution of the proposed method to the machine learning and representation learning communities, in terms of novelty, appears to be limited. Most of the involved methods in this work are mature DL techs. The primary novelty of this work lies in utilizing the untapped potential information from the raw signals. Given this focus, the manuscript would be more suited for  a bioinformatics-specific paper.

5. It is not demonstrated that whether the untapped potential information from the raw signals helps in clustering the sequences, which is claimed as the main contribution as in the Abstract. There should be a comparision between the raw-signal/DNA-sequence as inputs to support this claim.

6. The reviewer also concerns about the effectiveness of employing supervised classification tasks to derive embeddings,
particularly since approximating distance metrics is a relatively straightforward task with the datasets used in this study, which is also used in the existing works presented in item 1.

7. The author(s) may want to improve the writting. e.g.  the reviewer did not find any table or figure which is refered by "The results" in line 313.

**Questions:**

N/A

---

### Official Review · Reviewer_6ixx · 2024-10-30

**Soundness:** 4
**Presentation:** 3
**Contribution:** 4
**Rating:** 8
**Confidence:** 4

**Summary:**

The paper presents a deep learning-based approach for clustering DNA reads using raw electrical signals generated during nanopore sequencing. Traditionally, DNA clustering occurs after basecalling, where raw signals are converted into nucleotide sequences. The authors propose to bypass this step by clustering directly on the raw signals, which retain more information than the processed sequences.

The paper proposes the use of deep signal embedding to improve the clustering stage of DNA storage pipelines, bypassing traditional methods that rely on basecalled DNA reads. Experiments indicate that the proposed method outperforms state-of-the-art approaches.

**Strengths:**

- The core contribution is a model, based on the model in the Dorado basecaller, that performs clustering on these raw signals before basecalling. To my knowledge, this signal-based approach is novel and promising in the field of DNA storage.
- The authors evaluate their method on several DNA datasets. Their results show that signal-based clustering outperforms existing methods in terms of both time and accuracy.

**Weaknesses:**

Major:
- An important point that the authors should address is that it is uncertain whether the clustering gains translate into significant improvements in the final data retrieval phase.

Minor:
- Line 40: The description 'a "retrieval" stage where reads are decoded back to binary data files while correcting any errors using the chosen coding methods' is inconsistent with Figure 1, where the decoding is shown to occur after the retrieval stage.
- Line 80: "edits" should be replaced by "substitutions"
- Line 104: "contributions to" -> "contributions of"
- Line 156: How can UCLUST (published 2020) be based on USEARCH (published 2021)?
- Line 173: There are also algorithms such as GCTW (Zhou and De la Torre, 2016) for sequence alignment with sub-quadratic complexity. This should be acknowledged.
- Line 183: "during during" -> "during"
- Line 186: "maps" -> "map"
- Lines 189-193: This paragraph seems rather disorganized.
- Line 214 (footnote): "Deep DNA test" -> "Deep DNA Test", "Deep DNA pilot" -> "Deep DNA Pilot"
- Lines 199-206: This paragraph needs improvement. The text says that the synthesis will produce '"fast5" file formats'. Firstly, the synthesis yields oligonucleotides. Secondly, nanopore sequencing would yield raw signal data in the form of files in the FAST5 format.
- Line 223: I do not understand how to deduce from Table 1 that the strands are correctly clustered with the k=20 maximum distance constraint.
- Line 247: "Clmap" -> "Clamp"
- Line 250: The authors should be clearer about what is meant by the number of classes. It only becomes clear later that this refers to the number of (expected) clusters.
- Figure 2: The figure could be improved. On the left, it is not clear that the input files are used to synthesize the DNA. Also, according to the figure, it seems that the ArcFace loss is an input to the CNN.
- Line 409: "due to is due to" -> "due to"

**Questions:**

- Raw sequencing signals can vary considerably due to noise and machine-specific artifacts. How does the model handle this variability, and what measures are taken to ensure robust clustering across varying signal qualities?

---

### Official Review · Reviewer_KRVi · 2024-11-02

**Soundness:** 3
**Presentation:** 3
**Contribution:** 3
**Rating:** 5
**Confidence:** 3

**Summary:**

This paper proposes a novel approach to clustering in DNA data storage by focusing on clustering raw signals from Nanopore sequencing, rather than traditional basecalled DNA sequences (A/T/C/G). Traditional DNA storage pipelines typically perform clustering after basecalling, which introduces errors and computational inefficiencies. The authors aim to improve clustering accuracy and efficiency by directly embedding the raw signals produced during sequencing. The proposed deep learning model integrates the Dorado basecaller architecture and an ArcFace loss function, allowing for similarity-based grouping of raw signals. This approach is demonstrated to enhance clustering accuracy while reducing computational time by one to three orders of magnitude compared to traditional sequence-based methods. Results are validated on multiple datasets, and the model shows potential for enhancing both the clustering and reconstruction phases of DNA data storage systems.

**Strengths:**

Originality:
The paper addresses a novel problem by introducing raw signal clustering in DNA storage, which could set a new direction for handling sequencing data.

Relevance:
With data storage demands growing exponentially, improving DNA storage efficiency and accuracy is highly relevant.

Computational Efficiency:
The use of deep embeddings and cosine similarity yields significant computational improvements, making the method scalable and suitable for high-throughput sequencing.

Clarity in Experimental Validation:
Experimental results across multiple synthetic datasets support the authors’ claims, with performance metrics like AUC, ROC curves, and computation times well-documented.

**Weaknesses:**

Limited Generalizability Due to Custom Dataset:
The dataset is highly customized, relying on specific design files and synthetic DNA sequences generated by a particular synthesis provider (Twist Bioscience). Since real-world DNA samples often include much higher variability, especially in natural genomic data, the results from this dataset may not generalize well to other applications or to DNA data with biological origins rather than synthetic sources.

Fixed Threshold for Edit Distance (k=20):
The authors limit their edit distance calculations to a maximum of 20 differences, which helps with computational efficiency but may introduce inaccuracies. For reads that are more distant from their original design strands, this constraint could lead to incorrect cluster assignments, particularly if the dataset scales up to include a broader diversity of sequences with higher variances.

Different similarity comparisons:
The comparison between raw signal embedding using cosine similarity and DNA strands using edit distance may not be entirely fair due to their application to different data types—continuous versus discrete. Cosine similarity captures vector orientation in high-dimensional spaces, while edit distance measures specific sequence transformations. This discrepancy can lead to inconsistencies in capturing variations and information. Ensuring a fair comparison might require using a consistent metric or providing a rationale for the chosen methods.

May lack of novelty:
Aside from making and utilizing raw signals, the paper lacks additional innovation in either its model architecture, training strategies, or data processing. The use of dorado CRF layers and standard methods doesn't introduce novel techniques. To enhance innovation, further exploration in these areas would be beneficial.

**Questions:**

Generalizability of the Dataset:
How do you plan to address the potential lack of generalizability due to the customized dataset? Have you considered testing the method on natural genomic data to assess its applicability in real-world scenarios?

Edit Distance Threshold:
Could you elaborate on the decision to set the edit distance threshold at 20? How do you plan to ensure accuracy in clustering when dealing with reads that might have more variations? Can you provide an ablation study on the impact of choosing different edit distance thresholds on clustering accuracy?

Similarity Comparisons:
Can you provide a rationale for using different similarity measures for raw signals and DNA strands? How might you align these metrics to ensure consistency in comparison?

Model Architecture and Novelty:
Are there any future plans to innovate the model architecture or training strategies beyond using raw signals? What potential modifications could enhance the novelty of your approach?

---

### Official Review · Reviewer_2jVo · 2024-11-04

**Soundness:** 1
**Presentation:** 2
**Contribution:** 2
**Rating:** 3
**Confidence:** 4

**Summary:**

This paper tackles the clustering problem of DNA storage by using raw Nanopore signals as opposed to first translating the signals to strings (basecalling) and then clustering on the text strings. The motivation here is clear because the process to go from raw signals to strings can be very slow (weeks to months) and hence operating directly on the signals can be more efficient. Therefore, the authors design a deep learning method that operates on the data before it is discretized into bases. This paper presents a deep neural network that clusters these signals directly with reduced computation times compared to current approaches that cluster after basecalling.

From a technical point of view, the authors start with a pre-trained model and then do fine-tuning on additional data. The base model is Dorado, which is the standard basecaller for Nanopore reads. After truncating to get an embedding, they append a linear layer with output dimension equal to the number of classes (number of clusters). Then the training also uses the ArcFace loss function to increase separation between the clusters based on cosine similarity.

**Strengths:**

This paper studies an important problem in DNA storage. It has long been an open question whether systems can utilize information in the raw signals to better improve the analysis and downstream use of Nanopore data.

The authors show the effectiveness of taking a pre-trained model (Dorado) for Nanopore basecalling and repurposing it for a downstream application of the raw signals. This is pretty interesting because it is good to know the pre-trained model can be useful, and researchers do not have to start from scratch and train a model on Nanopore data themselves.

The authors also motivate their work by saying that the computation time can be much faster than basecalling with an existing algorithm. This is indeed an important aspect to improve upon. The methods in this paper could take the signal processing time from days/weeks down to under a single day, which would make DNA data storage more feasible for recovering stored data.

**Weaknesses:**

* A major weakness of the approach seems to be that the model needs to be trained with an output dimension equal to the number of clusters. However, the number of clusters in real settings can be very, very large. E.g., for a file that is stored that is 100MB, there could easily be over 1M clusters. I don't think network training will scale well in this case. So the experiments on 500 clusters are not representative of a real setting. Concretely, can the authors address how their approach might scale to much larger numbers of clusters (e.g. 1M+) that would be encountered in real-world applications. Please discuss potential solutions or modifications to their method to handle this challenge.

* In general, the experiments in Section 6 seem quite ad-hoc. The number of clusters is restricted without much discussion. One interesting outcome of this research could be how to train/design a good Nanopore signal clustering algorithm. But there are very few ablations, and not much details about what the authors learned or why their method works/fails. It feels as if the authors did one set of experiments and quickly wrote them up. This is below the bar for a ML paper at a top conference. There should be more tangible findings, and more justification for the design decisions. Specifically, can the authors investigate the impact of varying the number of clusters, explore different network architectures, and analyze how different components of their method contribute to its performance.

* The experimental results in Section 7 are missing comparisons to baselines. For example, the conclusion states that " Experiments reveal that the signal-model outperforms conventional clustering algorithms like Clover and Microsoft’s, providing faster and more accurate results." However, I don't see these experiments in the paper. For example, Table 2 and Figure 6 only seem to present results for the authors' algorithm, not the baselines (e.g., Clover or Microsoft's algorithms for clustering). It seems necessary to include direct comparisons with Clover, Microsoft's algorithm, and other relevant baselines in Table 2 and Figure 6, using the same datasets and evaluation metrics. This would provide a clearer picture of the proposed method's performance relative to existing approaches.

* At a high level, the related work is oddly about clustering, but the paper is about more than clustering. I am not familiar with things like how the Dorado method was trained, what competing methods there are, etc. Also have people really not tried to train deep learning algorithms on Nanopore data for other applications? This seems very surprising to me. For example, a quick search shows some other papers that use nanopore signals for applications:

  * HycDemux: a hybrid unsupervised approach for accurate barcoded sample demultiplexing in nanopore sequencing
Renmin Han, Junhai Qi, Yang Xue, Xiujuan Sun, Fa Zhang, Xin Gao & Guojun Li
Genome Biology volume 24, Article number: 222 (2023)

  * Kovaka S, Fan Y, Ni B, Timp W, Schatz MC. Targeted nanopore sequencing by real-time mapping of raw electrical signal with UNCALLED. Nat Biotechnol. 2021;39(4):431–41.

  * Overall, can the authors expand their discussion of related work to include: A more comprehensive overview of Nanopore signal processing methods, including Dorado and its competitors; A broader discussion of deep learning applications on Nanopore data beyond clustering; An analysis of how their approach compares to or builds upon these, as well as other relevant work in the field.

**Questions:**

* The claims about previous clustering algorithms do not seem very accurate or thorough. For example, the authors say "Unlike modern DNA clustering algorithms which are commonly used, such as Clover (Qu et al., 2022) and Microsoft’s (Rashtchian et al., 2017), the signal-model computation time is considerably faster." I am not sure how this is verified. For example, one inaccuracy is in Section 7.1 -- the authors say "Microsoft’s algorithm (Rashtchian et al., 2017) is orders of magnitude slower than Clover" but the cited paper GradHC (Ben Shabat, 2023) points out that the Microsoft algorithm is very fast, except when the clusters are very small and it does not converge (which may not be the case for the datasets in this paper, since the number of clusters is 50 in Section 6, and the number of reads is much larger). Also why not compare to the GradHC algorithm, which is shown to be quite fast?

* I am confused about "ground truth for clustering" and in particular the statement "Table 1 shows that for each of the experiments, k = 20 ensures each strand will be categorized to the correct cluster with high probability since the majority of the reads are closely aligned to their intended design" In Table 1, it seems that for the first two datasets, the average edit distance is between 52 and 63. So what happens for the strands with edit distance more than k = 20? Will they be assigned to a different cluster?

* Section 7.2 seems to be unfinished. There are no experiments to back up the claims. Also this sentence in 7.2 does not make sense as written: "In contrast, since signal-model directly uses the raw signals, the clustering algorithm will be given more data for the reconstruction algorithms." The clustering algorithm will operate on the reads, while the reconstruction algorithm will operate on the clusters. I don't know what it means for one algorithm to be given more data for another algorithm.

* One of the more interesting aspects to explore is whether other parts of the DNA storage system can be replaced with deep learning. For example, is it possible to train a model to go directly from raw signals to the final prediction for the stored data? Perhaps the network can learn to implicitly cluster then reconstruct then error correct, without needing separate algorithms for each of these steps.

* See also the questions at the end of each of the weaknesses above.

---

### Note · Authors · 2024-11-25

I have read and agree with the venue's withdrawal policy on behalf of myself and my co-authors.